# Analysis of ancestry heterozygosity suggests that hybrid incompatibilities in threespine stickleback are environment dependent

Ken A. Thompson[1]*, Catherine L. Peichel[2], Diana J. Rennison[3], Matthew D. McGee[4], Arianne Y. K. Albert[5], Timothy H. Vines[6], Anna K. Greenwood[7], Abigail R. Wark[8], Yaniv Brandvain[9], Molly Schumer[10,11], Dolph Schluter[1]

1 Department of Zoology & Biodiversity Research Centre, University of British Columbia, Canada, 2 Division of Evolutionary Ecology, Institute of Ecology and Evolution, University of Bern, Bern, Switzerland, 3 Division of Biological Sciences, University of California San Diego, San Diego, California, United States of America, 4 School of Biological Sciences, Monash University, Melbourne, Victoria, Australia, 5 Women's Health Research Institute, Vancouver, British Columbia, Canada, 6 DataSeer Research Data Services, Vancouver, British Columbia, Canada, 7 Sage Bionetworks, Seattle, Washington, United States of America, 8 Harvard Medical School, Cambridge, Massachusetts, United States of America, 9 Department of Plant and Microbial Biology, University of Minnesota, Saint Paul, Minnesota, United States of America, 10 Department of Biology, Stanford University, Stanford, California, United States of America, 11 Howard Hughes Medical Institute, Maryland, United States of America

* ken.thompson@zoology.ubc.ca

**Data Availability Statement:** All data and analysis code have been deposited in the Dryad repository: https://doi.org/10.5061/dryad.h18931zn3.

## Abstract

Hybrid incompatibilities occur when interactions between opposite ancestry alleles at different loci reduce the fitness of hybrids. Most work on incompatibilities has focused on those that are "intrinsic," meaning they affect viability and sterility in the laboratory. Theory predicts that ecological selection can also underlie hybrid incompatibilities, but tests of this hypothesis using sequence data are scarce. In this article, we compiled genetic data for $F_2$ hybrid crosses between divergent populations of threespine stickleback fish (*Gasterosteus aculeatus* L.) that were born and raised in either the field (seminatural experimental ponds) or the laboratory (aquaria). Because selection against incompatibilities results in elevated ancestry heterozygosity, we tested the prediction that ancestry heterozygosity will be higher in pond-raised fish compared to those raised in aquaria. We found that ancestry heterozygosity was elevated by approximately 3% in crosses raised in ponds compared to those raised in aquaria. Additional analyses support a phenotypic basis for incompatibility and suggest that environment-specific single-locus heterozygote advantage is not the cause of selection on ancestry heterozygosity. Our study provides evidence that, in stickleback, a coarse—albeit indirect—signal of environment-dependent hybrid incompatibility is reliably detectable and suggests that extrinsic incompatibilities can evolve before intrinsic incompatibilities.

## Introduction

Hybrid incompatibilities—interactions among divergent genetic loci that reduce the fitness of hybrids—are a key component of reproductive isolation between diverging lineages [1].

**Funding:** The authors received no specific funding for this work.

**Competing interests:** The authors have declared that no competing interests exist.

**Abbreviations:** GBS, genotyping by sequencing; QTL, quantitative trait locus; SNP, single nucleotide polymorphism.

Incompatibilities have been studied most intensively in the context of sterility and mortality, in part because phenotyping in the laboratory is reliable and because such incompatibilities can have simple genetic architectures involving few loci [2,3]. These sorts of interactions have come to be called "intrinsic" hybrid incompatibilities due to the fact that there are conflicts within the hybrid genome that are expected to impact hybrids in most environmental contexts (although note that the strength of selection against some intrinsic incompatibilities can vary across environments [4,5]). Studies have shown that the number of loci involved in intrinsic incompatibilities tends to increase with genetic divergence between the parent species [6–8] and that incompatibilities can be common throughout the genomes of isolated conspecific populations [9,10]. Collectively, evolutionary biologists have made substantial progress toward identifying generalities about the evolutionary genetics of intrinsic hybrid incompatibilities.

Ecological selection could underpin incompatibilities if particular allele combinations render hybrids unable to function in their ecological environment, such as in avoiding predators or capturing prey. Several recent studies have shown patterns consistent with this effect, wherein hybrids have "mismatched" trait combinations and reduced fitness as a result [11–13]. Such studies have successfully demonstrated that incompatibilities due to trait mismatch exist, but links to the underlying genetics have not been made. Perhaps the most significant barrier to progress in studying the genetics of "ecological" hybrid incompatibilities is the unique difficulty of detecting them. The ability to detect individual incompatibilities depends on the genetic architecture underlying traits—that is, whether quantitative trait loci (QTL) have small or large phenotypic effects [14,15]. For example, Arnegard and colleagues [11] found that combining divergent jaw traits together in $F_2$ threespine stickleback fish (*Gasterosteus aculeatus* L.) hybrids likely reduced their fitness because these traits interacted in a manner that reduced suction feeding performance. The interacting jaw components map to several regions of the genome that individually explain a small fraction (<10%) of the phenotypic variance (and most variance was unexplained), thus rendering it difficult to study their individual epistatic fitness effects.

Recent theoretical advances, however, suggest ways to test for and measure the net effect of hybrid incompatibilities using experimental crosses. Specifically, selection against hybrid incompatibilities in an $F_2$ hybrid cross causes an increase in ancestry heterozygosity—the number of sites in the genome that carry both parents' alleles at ancestry informative sites—at the population level [16,17]. This is expected because $F_2$ hybrids have a hybrid index of approximately 0.5—deriving half of their alleles from one parental species and half from the other. Thus, individuals with high heterozygosity relative to their hybrid index have fewer pairs of homozygous loci with opposite ancestry compared to relatively more homozygous individuals with a similar hybrid index. Assuming that most alleles affect the phenotype additively and have noninfinitesimal effect sizes, having many loci with opposite homozygous ancestry can result in hybrids with maladaptive "mismatched" phenotypes, whereas highly heterozygous individuals are expected to have less mismatched phenotypes (Fig 1, S1 Fig). Whether "mismatch" affects fitness, however, ultimately depends on the ecology of the system and the underlying fitness landscape. Such coarse approaches—coarse because they use summary statistics rather than direct mapping—are a promising means to identify the presence of small effect hybrid incompatibilities at the genetic level using field experiments.

In this study, we compare patterns of selection on ancestry heterozygosity between $F_2$ hybrid stickleback raised indoors in aquaria to those from the same cross types raised in the field in experimental ponds (see Table 1 for overview of data sources). We first consider hybridization between sympatric benthic and limnetic populations of threespine stickleback. These populations, which are reproductively isolated species due to strong assortative mating [19] and experience reduced hybrid fitness due to extrinsic selection pressures [20], have

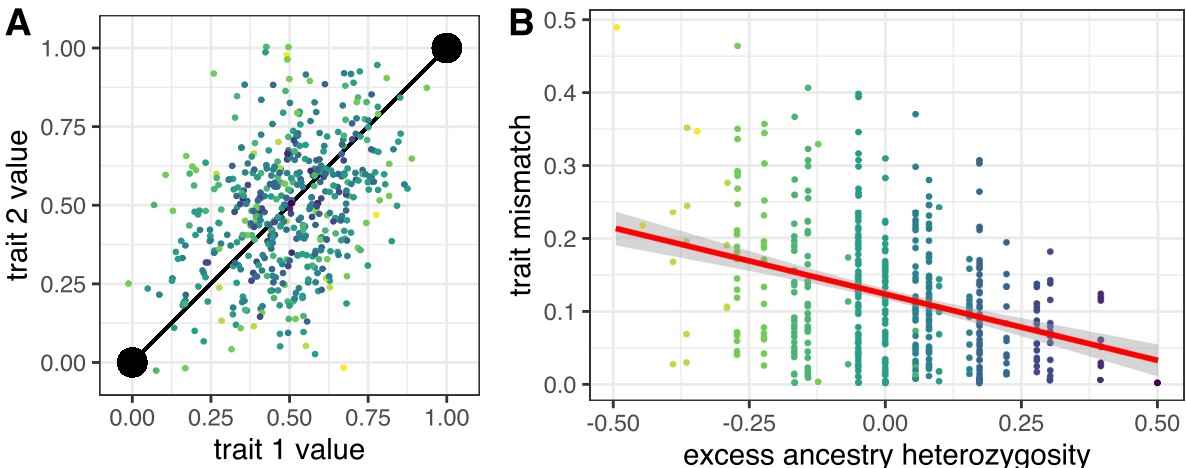

**Fig 1. Results from simulations illustrating how an ecological mechanism could underlie the heterozygosity–incompatibility relationship in F2 hybrids.** Both panels depict results from a representative simulation run of adaptive divergence and hybridization between 2 populations. We consider an organism with 2 traits that have both diverged as a result of selection. Colored points are individual hybrids, with darker colors indicating higher heterozygosity. Panel **(A)** depicts the distribution of 500 $F_2$ hybrid phenotypes in two-dimensional trait space. Large black points are the 2 parent phenotypes, which are connected by a black line indicating the "axis of divergence." Panel **(B)** depicts the relationship between individual excess ancestry heterozygosity and trait "mismatch" of individual hybrids [13]. Excess ancestry heterozygosity is the observed heterozygosity minus the expected heterozygosity based on ancestry proportion—0 is the expected mean in the absence of selection (approximately observed heterozygosity frequency of 0.5). Mismatch is calculated as the shortest (i.e., perpendicular) distance between a hybrid's phenotype and the black line connecting parents in (A). Variation parallel to this axis connecting parents in (A) captures variation in the "hybrid index." The plot shows that trait mismatch is lower in more heterozygous $F_2$ hybrids. Heterozygosity values are discrete because a small number of loci underlie adaptation in the plotted simulation run. Simulations are outlined in the Methods. The "mismatch"–heterozygosity relationship is stronger, although less intuitive, in organisms with greater dimensionality (i.e., more traits; see S1 Fig for a case with 10 traits following [18]). The data and code required to recreate this figure may be found at https://doi.org/10.5061/dryad.h18931zn3.

evolved independently in at least 5 watersheds in British Columbia, Canada [21,22]. Although reproductively isolated in the wild, the species pairs have no known intrinsic barriers that reduce fitness in the lab [20]. Second, we consider hybridization between allopatric populations of anadromous and solitary freshwater stickleback. As with the benthic–limnetic species pairs, these populations are recently diverged and can readily hybridize. The benthic × limnetic crosses are between specialist populations that are at extreme opposite ends of the range of trophic phenotypes observed among this region's native stickleback in fresh

**Table 1. Summary of data sources.**

| Cross type | Design | Study | Population | Method | Generation | Environment | *n* fish | *n* loci ± [1 SD] |
|---|---|---|---|---|---|---|---|---|
| Ben × lim | Biparental | unpublished | Priest | Microsatellites | $F_2$ | Lab | 90 | 22.9 ± 1.1 |
| Ben × lim | Biparental | Conte and colleagues [26] | Priest | SNP array | $F_2$ | Pond | 412 | 89.0 ± 0.0 |
| Ben × lim | Biparental | unpublished | Paxton | Microsatellites | $F_2$ | Lab | 89 | 97.1 ± 5.8 |
| Ben × lim | 8 × $F_0$ | Arnegard and colleagues [11] | Paxton | SNP array | $F_2$ | Pond | 615 | 62.5 ± 17.9 |
| Ben × lim | Biparental | Conte and colleagues [26] | Paxton | SNP array | $F_2$ | Pond | 636 | 62.0 ± 0.0 |
| Ben × lim | See methods | Bay and colleagues [27] | Paxton | SNP array | $F_2$ | Pond | 302 | 183.2 ± 19.7 |
| Ben × lim | 4 × biparental | Rennison and colleagues [28] | Paxton | GBS (RAD) | $F_2$ and $F_3$ | Pond | 649 | 85.1 ± 34.0 |
| Marine × fresh | Biparental | Rogers and colleagues [29] | LCR* × Cranby | Microsatellites | $F_2$ | Lab | 374 | 59.2 ± 4.3 |
| Marine × fresh | Biparental | Schluter and colleagues [30] | LCR* × Cranby | SNP array | $F_2$ and $F_3$ | Pond | 723 | 120.3 ± 5.6 |

*Little Campbell River anadromous.

GBS, genotyping by sequencing; RAD, restriction site–associated DNA; SNP, single nucleotide polymorphism.

water. The marine × freshwater cross involves a limnetic-like marine population and a generalist freshwater population (Cranby Lake) that is intermediate between the limnetic and benthic populations [23,24] and is therefore a less divergent cross with respect to trophic characters—however, the populations differ in other traits involved in marine–freshwater divergence [25].

If ecological selection on trait mismatch is operating in the field but not in the lab, selection for increased ancestry heterozygosity should be specific to the field (or at least stronger than in the lab). If mismatch is deleterious, the fitness landscape is assumed to be saddle or ridge like (see [11]); hybrids with mismatched phenotypes are displaced along the steep sides orthogonal to the axis of parental divergence and have lower fitness than individuals with relatively "matched" trait values (whether parental or somewhat intermediate). Thus, we predicted that we would observe elevated excess ancestry heterozygosity in samples from the field compared to those from the lab.

## Results

In support of our prediction, mean individual excess ancestry heterozygosity—the deviation from Hardy–Weinberg expectations based on the relative frequency of alternative ancestry alleles in the genome—was significantly elevated among pond-raised fish compared to aquarium-raised fish. This was the case in both the benthic × limnetic data ($\hat{\beta}$ = 0.021 ± 0.0081 [magnitude of excess ancestry heterozygosity ± SE], $z$ = 2.62, $P$ = 0.009) (Fig 2A—left; also see S2 Fig for plots of individual hybrid index and heterozygosity) and the marine × freshwater data (Fig 2A—right; $\hat{\beta}$ = 0.038 ± 0.0065 [SE], $z$ = 5.86, $P$ < 0.0001). Patterns were similar for all studies in the dataset—each study that contributed data from pond experiments found significant excess ancestry heterozygosity, and each study that contributed data from aquaria found that excess ancestry heterozygosity did not differ from 0 (Fig 2B). The signal of excess ancestry heterozygosity was variable among chromosomes, although the majority had values exceeding 0 (S3 Fig).

## Discussion

Ecological selection acting on hybrids is a critical determinant of gene flow between diverging lineages [33]. Yet, detecting how divergent alleles interact to mediate hybrid fitness in ecological contexts has proven difficult due to the effect sizes of interacting loci and the massive experiments required to achieve sufficient power [34]. Here, we tested whether a coarse-grained signal of selection against hybrid incompatibilities—elevated excess ancestry heterozygosity [17]—differed between laboratory and field replicates of genetic crosses between the same populations. We found that excess ancestry heterozygosity was elevated in recombinant stickleback hybrids raised in experimental ponds compared to those from similar crosses raised in aquaria. This result is consistent with the hypothesis that certain ecologically mediated hybrid incompatibilities between recently diverged stickleback populations act more strongly in field settings than in the lab. Our finding implies that individual stickleback with a greater mismatch in parental traits are less likely to survive than those with lesser mismatch. Below, we consider whether other processes could plausibly explain this result, discuss the relevance of our findings for speciation, and highlight opportunities for future research.

The lack of excess ancestry heterozygosity in hybrids raised in aquaria is an expected result given what is known about "intrinsic" hybrid incompatibilities in stickleback. Previous studies of benthic × limnetic hybrids have found no evidence for intrinsic inviability in $F_2$ crosses using measures of embryo development and hatching success [22,20,35] or lifetime fitness [20]. A recent review summarizing the literature on reproductive isolation in threespine stickleback [36] reports that "intrinsic" barriers are typically weak to nonexistent. Both marine (i.e.,

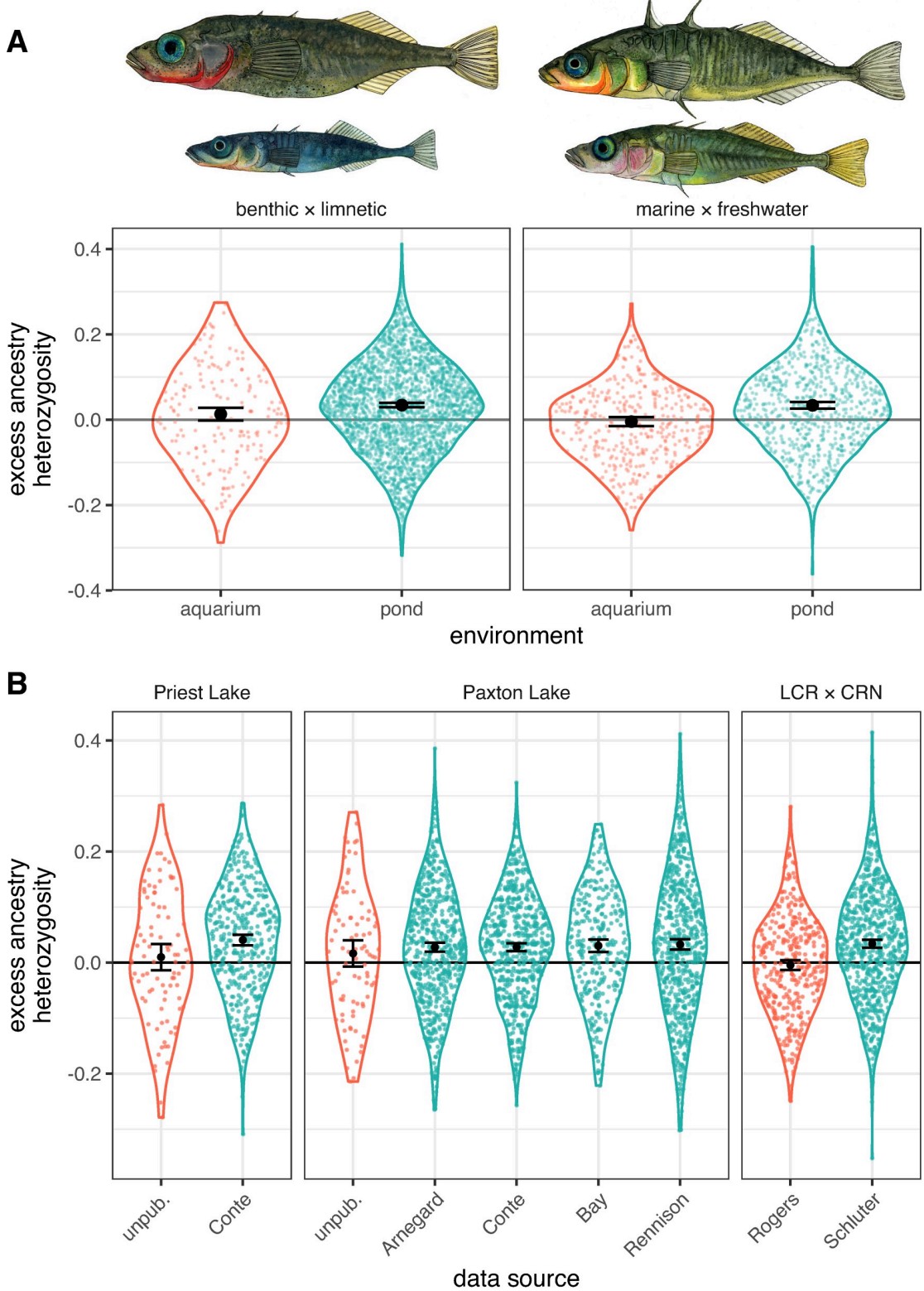

**Fig 2. Excess ancestry heterozygosity in recombinant threespine stickleback hybrids in the lab (aquarium) and field (pond).** Panel (**A**) shows the main test of group differences for (left) benthic × limnetic crosses and (right) marine × freshwater crosses. Drawings above the panels were done by K. Chu and show the first listed population on top. These values are extracted from our statistical model using visreg (colored points and violins; [31]) and emmeans (black estimates of means and CIs; [32]). Panel (**B**)

shows raw data (i.e., not from a statistical model) for each data source (Table 1; LCR × CRN is Little Campbell River × Marine) separately, with colors representing lab (red) versus pond (blue) as in (A). Violin overlays show the full distribution of the data, and small colored points show values for individual fish. Large black points are the group means and 95% CIs. The data and code required to recreate this figure may be found at https://doi.org/10.5061/dryad.h18931zn3.

anadromous) × freshwater and benthic × limnetic crosses had no evidence for intrinsic inviability, whereas the authors found evidence for hybrid ecological inviability in both systems [36]. Thus, it would have been surprising if we had recovered any signal of selection on excess ancestry heterozygosity in lab-raised hybrids.

## Alternative causes of excess ancestry heterozygosity

We hypothesize that selection against trait mismatch (i.e., ecological hybrid incompatibility) caused the observed patterns of selection on ancestry heterozygosity among surviving individuals in the ponds, but several other mechanisms could possibly underlie such a pattern. These alternative mechanisms involve processes operating at single loci rather than interactions among loci. Ultimately, the data presented here have limited ability to conclusively distinguish between single-locus processes like heterosis and multilocus processes like incompatibilities, but we discuss the strength of evidence for different possible alternative mechanisms below.

Heterosis refers to a case where, for a given locus, the heterozygote has greater fitness than the parental genotypes. This could obviously lead to selection for elevated ancestry heterozygosity. In the benthic × limnetic crosses, environment-specific heterosis is unlikely based on prior knowledge about hybrid fitness in this system. If heterosis were common, then $F_1$ hybrids should have higher fitness than parents. However, $F_1$ and reciprocal backcross hybrids have lower growth and/or survival than both parent taxa in field experiments [20,37,38]. These patterns are opposite to what would be expected as a result of field-specific heterosis, suggesting that it is unlikely to be acting in the benthic × limnetic crosses. Less is known about heterosis in marine × freshwater crosses, although Hagen [39] found evidence for strong ecology-based selection against hybrids in the hybrid zone between the Little Campbell River anadromous population and the resident freshwater-adapted stream population. In sum, the available evidence from observing selection on $F_1$ hybrids in the field versus lab is inconsistent with there being environment-specific heterozygote advantage. The above patterns also suggest that alleles conferring a fitness advantage are not dominant to lower fitness alleles and thus that environment-specific dominance for fitness does not result in pond-specific selection for increased ancestry heterozygosity.

It is conceivable that heterosis could act in the lab via body condition and general health, but only affects mortality in the field and thus would not affect mean excess ancestry heterozygosity in aquaria. Previous studies have found that the growth rate of $F_1$ benthic × limnetic hybrids in the lab matches the additive expectation of parents [20,40], consistent with there being no intrinsic hybrid vigor. In the present study, we find no relationship between body size and excess ancestry heterozygosity in any of the aquarium-raised crosses (S4 Fig; robust within-family regression analyses are not possible in ponds due to the small number of individuals within families; see [11] for discussion of using body size as a component of fitness). Thus, we conclude that there is no evidence supporting a relationship between ancestry heterozygosity and vigor in the laboratory.

We can also use predictions that are specific to the genetics of heterosis versus hybrid incompatibilities to differentiate between them. Specifically, heterosis depends on interactions within a locus, while incompatibilities depend on interactions between loci. If the benefit of ancestry heterozygosity was due to interactions within loci alone, we would expect to see no

relationship between genome-wide admixture proportion (i.e., hybrid index) and excess ancestry heterozygosity—this is because all heterozygosity is beneficial regardless of the genetic background. By contrast, if excess heterozygosity was attributable to interactions among loci, as in the hybrid incompatibility model, we would expect diminishing benefits of excess ancestry heterozygosity as genome-wide ancestry proportions became more parent like (i.e., as hybrid index deviates from 0.5). As expected under the incompatibility model, excess ancestry heterozygosity declines as the hybrid index of pond-raised individuals deviates from 0.5 (Spearman $\rho = -0.059$; $P = 0.0006$), while there is no relationship in the lab ($\rho = -0.012$; $P = 0.77$) (S5 Fig). While this observation does not eliminate the possibility that some of the observed excess heterozygosity is driven by environment-specific heterosis, it is consistent with our hypothesis that excess ancestry heterozygosity largely results from selection against extrinsic incompatibilities.

Inbreeding depression could also generate our observed pattern if it only affected fitness in ponds [41–43]. If inbreeding depression led to selection on ancestry heterozygosity in our study, we would expect inbred individuals to have higher mean excess ancestry heterozygosity than outbred individuals. To test this specific prediction, we used the data from Arnegard and colleagues [11] where due to the study design, some $F_2$ hybrids resulted from mating between full siblings ($n = 74$), and some individuals were produced by mating between unrelated individuals ($n = 541$). We find that these 2 groups do not differ in mean excess ancestry heterozygosity ($F_{1,612} = 0.0734$, $P = 0.787$; S6 Fig). Furthermore, the direction of the nonsignificant difference in mean (estimate of outbred − sibling difference = 0.0035 ± 0.0129 [SE]) is the opposite of what is expected under environment-specific inbreeding depression. We note, however, that this analysis does not account for the possibility of any fixed deleterious alleles in the populations [44,45]. Thus, the data suggest that environment-specific inbreeding depression is not causing heterosis in this system.

Finally, an additional analysis indirectly provides evidence of phenotype-based hybrid incompatibilities in the data from Arnegard and colleagues [11]. Specifically, Arnegard and colleagues [11] classified $F_2$ hybrids into 4 groups ("A," "B," "L," and "O") based on individual niche use. "B," "O," and "L" individuals had benthic-like, intermediate, and limnetic-like diets, respectively. "A" individuals, however, had unusual diets, were smaller, and had a greater extent of mismatched trait combinations compared to the other groups. The authors hypothesize that trait mismatch caused these fish to grow more slowly than more "matched" individuals. Our analysis reveals that "A" group individuals have lower excess ancestry heterozygosity than non-"A" individuals (S7 Fig)—as expected if lower excess ancestry heterozygosity correlates with higher trait mismatch. This reanalysis suggests a link between trait mismatch, ancestry heterozygosity, and fitness in stickleback hybrids raised in a natural environment.

## Relation to other studies of incompatibilities and ancestry heterozygosity

Our results contribute to a growing understanding of the biology of environment-dependent hybrid incompatibilities. In natural hybrid populations of swordtail fishes, Schumer and colleagues [46] estimated that dozens of incompatibilities separate parent species [47]. The authors also suggested that many of these are likely subject to natural or sexual selection [46]. Previous studies on hybrid stickleback [11,48] have estimated fitness landscapes that are consistent with the hypothesis that mismatched trait combinations are selected against, and our analysis of genetic data supports this hypothesis. We also note that selection against mismatched combinations of traits has the same genetic basis as selection against single phenotypes that express maladaptive transgressive values after hybridization [18,49,50]. Thus, studies focused on only a single trait under stabilizing selection might still find selection

against incompatibilities if hybrids have trait values that are below or above the optimum value [51]. More broadly, our results are consistent with predictions generated from theoretical models of speciation and adaptation [17]. Thus, although stickleback is an excellent system in which to test these predictions, the mechanisms underlying our results are likely general.

Our findings also highlight differences from previous analyses of selection on hybrids in different environments. In yeast, selection for low ancestry heterozygosity is common in hybrids when tested in the lab [52,53]. This difference (which does not occur as a result of aneuploidy) between the yeast studies and our study of stickleback might result from the fact the lab media that yeast were raised in are novel environments, and transgressive traits suited to these environments result from excess ancestry homozygosity (see S1 Fig; also see [54] for how novel environments can result in directional selection on ancestry). Stickleback populations in postglacial lakes are specialists on zooplankton or benthic invertebrates when they coexist with a competing fish species [23,55,56] or are generalist populations that make use of both niches [57]. Freshwater stickleback populations in this region span a range of phenotypes along a limnetic–benthic axis, and no specialists along novel trophic axes are known to occur. Fish in the experimental ponds that we considered herein have diets that are largely representative of what they consume in nature [11], and there is no evidence supporting the hypothesis that ponds contain novel adaptive peaks. Qualitative patterns of selection against hybrids driving excess ancestry heterozygosity might therefore depend on the availability and nature of novel ecological niches.

## Outlook, caveats, and conclusions

While we identify genetic signatures consistent with the existence of environment-specific hybrid incompatibilities, we cannot begin to identify their specific mechanisms, including when they arise during ontogeny, without connecting phenotype to genotype. Experiments that directly manipulate individual phenotypes, or manipulate interactions between individuals and their environments, are needed to establish such causality. As a field, we should aim to identify the types of traits that typically underlie ecological hybrid incompatibilities. Integrating field studies of hybrid incompatibility with QTL mapping of ecologically important traits [11,58] represents an exciting new frontier for empirical research into the mechanisms of speciation. Moreover, our data only allow us to scratch the surface of how incompatibilities are spread across the genome—it will be valuable for future studies with higher resolution genomic and phenotypic data to investigate this further.

We also do not know the strength of selection against ecological hybrid incompatibilities. Simple simulations (included in archived R scripts) illustrate that the strength of selection necessary to generate 3% excess ancestry heterozygosity in a population of $F_{FV}$ hybrids (similar to that observed in the present study) can vary by orders of magnitude depending on assumptions about the genetic architecture of selection. Moreover, the degree to which mismatched trait combinations are expressed in hybrids, and thus the ability to detect the coarse signal of ecological incompatibilities in an $F_2$ cross, depends considerably on the underlying genetic architecture of adaptive divergence—in particular, the number, effect sizes, and dominance of QTL. Because the expression of maladaptive trait combinations is predicted to increase with the magnitude of divergence between parent populations [18,24,50], we may predict that the strength of selection against ecological incompatibilities will increase with the magnitude of divergence between parents. However, quantifying the ecological basis of incompatibilities and their genetic structure will remain technically challenging.

The evidence presented here is consistent with the hypothesis that extrinsic hybrid incompatibilities are an important mechanism of postzygotic isolation in this system. Our results

imply that selection against ecologically mediated hybrid incompatibilities is active from the earliest stages of divergence. Speciation is largely complete when divergent lineages can stably coexist in sympatry, as is the case for the benthic and limnetic stickleback species pairs. Reproductive isolation between them is primarily thought to have arisen incidentally as a by-product of phenotypic divergence [20,38,59,60], with additional selection favoring the reinforcement of premating barriers as a result of low hybrid fitness [20,61]. Thus, our results are consistent with the idea that selection against trait mismatch, or ecological hybrid incompatibilities, is associated with extrinsic postzygotic isolation in this classic system of rapid and recent adaptive radiation.

## Materials and methods

### Data sources

We used both previously published and unpublished data in our analyses. Summary information about each data source is listed in Table 1. We base our main inference on a comparison of ancestry heterozygosity in hybrids born and raised in aquaria to hybrids from the same cross types born and raised in experimental ponds, which are seminatural ecosystems. See [11,30] for additional information about ponds. Pond-raised crosses capture both "intrinsic" and "extrinsic" incompatibilities, whereas aquarium-raised crosses are expected to capture "intrinsic" incompatibilities that impact hybrid fitness. In most cases, $F_2$ hybrids were produced via mating between full siblings in both the lab and field.

Studies raising fish from the same cross type (benthic × limnetic or marine × freshwater) in the same environment (aquaria or pond) were combined for analysis (see Fig 2). Two studies [28,30] genotyped both $F_2$ and $F_3$ hybrids, which we analyze together because we found that this choice does not affect our conclusions (S8 Fig). Similarly, data were analyzed together for 4 studies of benthic × limnetic hybrids from Paxton Lake raised in experimental ponds because excess ancestry heterozygosity was statistically indistinguishable among them ($F_{3,2217} = 0.304$; $P = 0.82$); note that this indicates that potential problems with genotyping by sequencing (GBS), such as improperly inferring heterozygotes [62], likely do not affect our conclusions. This grouping of studies was done only to simplify the presentation of results—patterns are highly repeatable across replicates, and analyses showing results for each pond and/or study separately are shown in S9 Fig. Relevant details of each data source are outlined below, but see the original studies for full details including animal use permits.

Studies genotyped fish using either microsatellites, single nucleotide polymorphism (SNP) arrays, or GBS. All 3 lab studies used microsatellites, whereas the pond studies all used SNP arrays or GBS. We examined potential concerns resulting from different genotyping methods and found no evidence that our results are caused by such differences. First, we only consider loci where parents have no alleles in common and thus can accurately polarize ancestry. Second, we only use loci that were heterozygous for ancestry in $F_1$s, so loci with "null" microsatellites or any difficulties in distinguishing alleles would be filtered out (see section on data filtering below). In the largest microsatellite dataset [29], 100% of loci that were different in parents were accurately called as heterozygous across 8 $F_1$s (288 of 288 loci across all 8 $F_1$ fish). SNP genotypes of the same cross type similarly had 100% heterozygosity in $F_1$s [30]. Finally, in 1,000 simulations resampling our dataset to only a single marker per chromosome, 99.4% of estimates of our statistical main effect were in the same direction as detected in the full dataset. In light of the above, we suggest our the differences detected between lab and pond datasets reflect biology rather than methodology. Allele frequencies and heterozygosity are shown for all genotyped loci (within a given dataset) in S10 Fig.

One additional difference between the pond and aquarium studies is that pond $F_2$ hybrids were a result of natural mating among $F_1$ hybrids, whereas aquarium $F_2$s were produced via artificial crosses. We do not anticipate that this will affect our conclusions, however, because we only consider loci that were fixed differences between $F_0$s, and thus are expected to segregate in a 1:2:1 pattern regardless of the process that united eggs and sperm.

## Benthic × limnetic crosses

We obtained data from 4 sources for the pond-raised benthic × limnetic hybrids. The data from aquaria are unpublished. Relevant details of each data source are given below. Final sample sizes from each data source are given in Table 1.

Conte and colleagues [26] generated a single $F_1$ family from each of the Priest and Paxton Lake species pairs. Both were founded by a single wild benthic female and a limnetic male that were collected and crossed in 2009. Thirty-five adult $F_1$ Paxton Lake hybrids and 25 adult $F_1$ Priest Lake hybrids were released into separate ponds where they bred naturally to produce $F_2$ hybrids. $F_2$ adults were collected over 1 year later and were genotyped using a SNP array [63]. A total of 246 SNPs were found in the Paxton cross, and 318 were found in the Priest cross.

Arnegard and colleagues [11] conducted a pond experiment with 8 $F_0$ grandparents from Paxton Lake. Two crosses were between limnetic females and benthic males, and 2 crosses were between benthic females and limnetic males. Five $F_1$ males and 5 $F_1$ females from each family were added to a single pond in March 2008 where they bred naturally. Juvenile $F_2$s were collected in October of that same year and genotyped at 408 SNPs using the SNP array.

Bay and colleagues [27] genotyped $F_2$ hybrid females between Paxton Lake benthics and limnetics. Fish are from several crosses and study designs. One used a cross with 4 unique $F_0$s that were used to produce 2 $F_1$ families—one with a limnetic as dam and the other with a benthic as dam. A second had 8 unique $F_0$s, where 2 $F_1$ crosses were in each direction. These 2 crosses used wild fish collected in 2007. A third set of crosses was done in 2009, 1 in each direction, then the 2 $F_1$ families were released into separate ponds. Since the goal of the authors' study was to examine the genetics of mate choice, a large number of $F_2$ females were genotyped at a small number of microsatellite markers. A subset of $F_2$ females identified in this parentage analysis were genotyped at 494 SNP markers using the SNP array. A total of 302 females were assigned to families with 10 or more full sibs (which was necessary for linkage mapping).

Finally, Rennison and colleagues [28] conducted a study with 4 unique Paxton Lake benthic × limnetic $F_1$ hybrid families that were each split between 2 ponds. One pond in each pair contained a cutthroat trout predator (ancestry heterozygosity did not differ across pond types and data are pooled across all pairs and pond types). Wild fish were caught in 2011, and $F_1$s were released in 2012. Fish bred naturally and juvenile $F_2$s were sampled in September of that same year. $F_3$ hybrids were collected in September 2013. Approximately 50 fish from each pond and hybrid generation were genotyped at over 70,000 loci using restriction site–associated DNA sequencing (GBS), and after filtering and selection of diagnostic loci, we retained 2,243 SNPs.

The Paxton and Priest Lake laboratory cross data are original to this study. Crosses used a single wild-caught benthic female fish and a single wild-caught limnetic male fish as $F_0$ progenitors. Wild fish were crossed in 2003. Sibling mating of $F_1$ hybrids was used to produce a single $F_2$ hybrid family for analysis, and fish were raised in glass aquaria and fed ad libitum. A total of 92 $F_2$s from the Priest Lake cross were genotyped at 84 microsatellite markers, and 86 $F_2$s from the Paxton Lake cross were genotyped at 216 microsatellite markers following [64]. We constructed a combined linkage map between the 2 families using the map integration function in JoinMap 3.0 [65]. The average centimorgan (cM) distance between the markers in

the Priest Lake and Paxton Lake maps was 15.6 ± 1.85 and 4.5 ± 0.51 (mean ± 1 SE), respectively. Use of animals was approved by UBC's Animal Care Committee (A97-0298).

## Marine × freshwater crosses

Schluter and colleagues [30] conducted a pond experiment with anadromous (hereafter "marine") × freshwater hybrids. This study crossed a marine female from the Little Campbell River, British Columbia, with a freshwater male from Cranby Lake, British Columbia. Over 600 juvenile $F_2$ hybrids were introduced into the ponds directly in August 2006. $F_2$s were produced from 6 $F_1$ families—6 unique females were crossed with 4 males (2 males were crossed twice each). $F_2$s overwintered with an estimated over winter survival rate of approximately 86% (from mark–recapture). In spring 2007, surviving $F_2$s bred and were genotyped at 1,294 bi-allelic SNP markers using a SNP array. A total of 500 of their $F_3$ hybrid offspring were collected in October 2007 and were genotyped with the same methodology.

The data for the laboratory marine × freshwater cross were originally published by Rogers and colleagues [29]. The population used a single Little Campbell River female as the $F_0$ dam and a single Cranby Lake male as the $F_0$ sire. Wild adult fish were captured in 2001 to generate a single $F_1$ hybrid family. Four $F_2$ crosses were made from individuals from this $F_1$ family (8 $F_1$ parents total). $F_2$ hybrid fish were genotyped at 96 microsatellite markers.

## Marker filtering and estimating excess ancestry heterozygosity

For each dataset, we restricted our analysis to loci where the $F_0$ progenitors of a given $F_2$ family had no alleles in common (e.g., all "BB" in benthic $F_0$s and all "LL" in limnetic $F_0$s) and where all $F_1$ hybrids were heterozygous for ancestry (e.g., all "BL"). GBS data were filtered to include SNPs with > 20× coverage for a given individual. Final sample sizes of fish and markers are given in Table 1. In all cases, the sex chromosome (chromosome 19) was not analyzed.

Because some studies have more individuals than loci, and others have more loci than individuals, we analyze ancestry heterozygosity both in individuals (averaged across loci) and at loci (averaged across individuals). We retained individuals for which at least 20 loci were genotyped, and retained loci for which at least 20 individuals were genotyped. Differences in genotyping success and/or family structure caused the number of genotyped loci to differ among individuals for a given study (Table 1). For simplicity, we focus on the analysis of individuals in the main text.

Deviation from the expected 50:50 ancestry proportions in $F_2$ hybrids—via variance introduced by the assortment of chromosomes into gametes, recombination, and/or directional selection against one ancestry—reduce the expected heterozygosity below 0.5. To account for this, we base our main inference on estimates of excess ancestry heterozygosity. Excess ancestry heterozygosity was calculated as observed ancestry heterozygosity ($p_{AB}$) minus expected ancestry heterozygosity ($2p_A p_B$, where $p_A$ and $p_B$ are the frequencies of both ancestries at the locus or in the individual's genome). Our conclusions are unchanged, however, if an uncorrected "observed" heterozygosity is used as the response variable (see S11 Fig) or if the expected heterozygosity is adjusted to account for sample size (i.e., multiplying by $\frac{2N}{2N-1}$ following [66]; both analyses included in archived R script).

## Data analysis

All data processing and analysis was done in R v4.1.1 [67] with packages included in the tidyverse [68]. We compared excess ancestry heterozygosity between aquarium and pond studies using a linear mixed model [69] where excess ancestry heterozygosity was the response, environment (aquarium or pond) cross (B×L or M×F), their 2-way interaction, and lake (Paxton

or Priest; for B×L crosses) were fixed factors, and data source was a random effect. We note that our conclusions are unchanged if we analyze excess ancestry heterozygosity of loci averaged across individuals, rather than ofindividuals averaged across their genotyped loci (S12 Fig). We used the emmeans [32] package to evaluate the statistical significance of between-group differences and visreg [31] to visualize models and extract residuals.

Another possible cause of excess ancestry heterozygosity is genotyping error. Simulations of genotyping error—where all errors are conservatively assumed to have resulted in true homozygotes being called as heterozygotes—indicate that error rates in excess of 5% are necessary to cause the pattern we observe (see archived R script). We believe that our error rate is much smaller than 5%, because we encountered no cases in which a locus was falsely called as homozygous in $F_1$ hybrid individuals (which, barring a rare mutation, are known to be heterozygous in all ancestry informative markers). In addition, we expect that genotyping errors would affect lab and pond data similarly.

## Simulations underlying conceptual figures

We used simple simulations in Fig 1 of the main text to illustrate the mechanistic relationship between trait mismatch and heterozygosity. Similar results have been noted elsewhere [17,18], but we give our detailed methods herein. We consider the following life history: A single population adapts to a novel environment and then hybridizes with the ancestral population. The phenotype and genotype (hybrid index and heterozygosity) are recorded for $F_2$ hybrids.

We use the framework of Fisher geometric model [70], wherein the phenotype of an organism is a vector of $m$ traits, $\mathbf{z} = [z_1, z_2 \ldots, z_m]$. For simplicity and ease of visualization, we only consider $m = 2$ in the main text (see S1 Fig for $m = 10$). We assume that mutations are sufficiently rare that they individually sweep through an otherwise monomorphic population. We also assume that mutations all occur at unique loci with free recombination among them (i.e., no linkage) [71,72]. Mutations influence the phenotype additively and are vectors of length $m$ where values are drawn from a random normal distribution with a mean of 0 and a standard deviation of $\alpha$ ($\alpha = 0.15$). The fitness of a given population is calculated as $w = \exp(-\sigma \|\mathbf{z}-\mathbf{o}\|^2)$, where $\|\mathbf{z}-\mathbf{o}\|$ is the Euclidean distance between the populations current phenotype ($\mathbf{z}$) and the optimum ($\mathbf{o}$), and sigma is the strength of selection ($\sigma = 10$). The original phenotype of the population is $z_0 = [0,0]$, and the optimum phenotype is $z_{opt} = [1,1]$. The selection coefficient $s$ value when they arise, where $s = w_{mut}w_{wt}-1$. The probability that a given mutation fixes, $\pi$, is calculated as $\pi = 1-\exp(-2Nsp)/1-\exp(-2Ns)$, where $N$ is the effective population size ($N = 1,000$), $p$ is the frequency of the mutation in the population ($p = 1N$), and $s$ is the selection coefficient.

We allow 1,000 mutations to arise in our simulations, which is sufficient for the adapting population to reach the optimum. After adaptation, we simulate hybridization with the ancestral population (which contains no mutations and has a value of 0 for both traits). We generate 500 $F_2$ hybrids, which inherit homozygous ancestral, heterozygous, or homozygous derived ancestry at each locus with probabilities 0.25:0.5:0.25. The genotype of these hybrids determines their phenotype for both traits; these phenotypes are plotted in Fig 1A. We consider 2 orthogonal properties of the phenotype: hybrid index and heterozygosity. Hybrid index is the fraction of alleles an individual inherited from the derived parent, and heterozygosity is the fraction of loci that are heterozygous. We finally calculated the Euclidean phenotypic distance from each hybrid's phenotype to the line connecting parent phenotypes. This distance is the individual hybrid's "mismatch" as calculated elsewhere [13].

All data and analysis code have been deposited in the Dryad repository: https://doi.org/10.5061/dryad.h18931zn3 [73].

## Supporting information

**S1 Fig. Simulation model illustrating the incompatibility–heterozygosity relationship.** The model is as in Fig 1 in the main text except there are 10 traits instead of 2 and the optimum of the adapting population is "0" for traits 2 to 9. Plots and model are inspired by Fig 1 in [18]. Both panels depict results from a representative simulation run of adaptive divergence and hybridization between 2 populations. Colored points are individual hybrids, with darker colors indicating higher heterozygosity. The left panel depicts the distribution of 500 $F_2$ hybrid phenotypes where the x-axis depicts the value of the selected trait, and the y-axis depicts the Euclidean distance from the optimum for traits 2–9 (i.e., $y = \sqrt{\sum_{i=2}^{10} z_i^2}$). Large black points are the 2 parent phenotypes. The right panel depicts the relationship between excess ancestry heterozygosity and the maladaptive distance from the optimum for individual hybrids [13]. Points are slightly jittered horizontally. The plot shows that this maladaptive trait expression is lower in $F_2$s with greater excess ancestry heterozygosity. Heterozygosity values are fairly discrete because a small number of loci underlie adaptation in the plotted simulation run. The data and code required to recreate this figure may be found at https://doi.org/10.5061/dryad.h18931zn3.
(PDF)

**S2 Fig. de Finetti ternary diagrams for genotyped individuals.** Each point represents an individual hybrid and shows each individual's hybrid index (frequency of benthic or marine alleles in its genome) and its mean heterozygosity. Hybrid index and heterozygosity are used because many loci are being considered simultaneously. These graphs are not used for analysis, but rather are shown to allow readers to visualize the structure of the raw data that underlies our analysis. Specifically, the shapes of the distributions of heterozygosity and hybrid index values are similar between environments and crosses—the means are just subtly different. The data and code required to recreate this figure may be found at https://doi.org/10.5061/dryad.h18931zn3.
(PDF)

**S3 Fig. Mean excess ancestry heterozygosity across chromosomes.** Each point is the average excess ancestry heterozygosity for all loci on a given chromosome (raw values; not residuals from a statistical model). Linkage group XIX contains the sex determining region and is plotted or analyzed. Error bars are 1 SE. The data and code required to recreate this figure may be found at https://doi.org/10.5061/dryad.h18931zn3.
(PDF)

**S4 Fig. No relationship between individual mean heterozygosity and growth (standard length) in the aquarium-raised biparental benthic–limnetic $F_2$ hybrids.** Results are residuals from visreg [31]. Each point is an individual $F_2$ hybrid. Standard length is standardized within family (1 family each for Paxton and Priest lakes for B×L and 4 families for M×F. The interaction between lake-of-origin mean heterozygosity was nonsignificant so we plot the main effect across both lakes of origin (Paxton and Priest). Mean heterozygosity was not significantly associated with standard length for either cross ($B \times L - \hat{\beta} = -0.05 \pm 0.67$ [SE], $F_{1,174} = 0.0075$, $P = 0.93$; $M \times F - \hat{\beta} = 0.65 \pm 0.59$ [SE], $F_{1,372} = 1.24$, $P = 0.26$). Analyses considering body depth (either individually or in a combined metric of "overall size") give the same qualitative result (see archived R script). The data and code required to recreate this figure may be found at https://doi.org/10.5061/dryad.h18931zn3.
(PDF)

**S5 Fig. The benefit of excess heterozygosity declines with deviations from a hybrid index of 0.5 in ponds but not in the lab.** Each point is an individual recombinant hybrid and data are pooled across cross types. Excess ancestry heterozygosity declines as the hybrid index of pond-raised individuals deviates from 0.5 (Spearman $\rho = -0.059$; $P = 0.0006$), whereas there is no relationship in the lab ($\rho = -0.012$; $P = 0.77$). Bootstrap tests indicate that these 2 correlations are statistically indistinguishable, so we consider this analysis to be interesting and consistent with our hypothesis, but not conclusive. The data and code required to recreate this figure may be found at https://doi.org/10.5061/dryad.h18931zn3.
(PDF)

**S6 Fig. Comparing selection on ancestry heterozygosity between individuals of different relatedness.** There is no difference in mean excess ancestry heterozygosity between $F_2$ hybrids whose parents were unrelated and those whose parents were full siblings. Data from [11]. The data and code required to recreate this figure may be found at https://doi.org/10.5061/dryad.h18931zn.
(PDF)

**S7 Fig. Fish assigned a priori as "phenotypically mismatched" ("A" group) have lower excess ancestry heterozygosity than nonmismatched fish.** Assignments are from [11] and methods are described therein. Each point is an individual $F_2$ hybrid. This result implies that phenotypically mismatched individuals have lower excess ancestry heterozygosity than nonmismatched individuals. The data and code required to recreate this figure may be found at https://doi.org/10.5061/dryad.h18931zn3.
(PDF)

**S8 Fig. Excess ancestry heterozygosity does not differ between $F_2$ and $F_3$ hybrids.** The plots show individual excess ancestry heterozygosity from the 2 studies that genotyped both the $F_2$ and $F_3$ generations [28,30]. The means (black dots, ± 95% CI) do not differ between generations in either study (Rennison group difference = 0.014 ±0.0095 [SE], $F_{1,667} = 2.35$, $P = 0.13$; Schluter group difference = 0.0016 ±0.0078 [SE], $F_{1,721} = 0.042$, $P = 0.84$). The data and code required to recreate this figure may be found at https://doi.org/10.5061/dryad.h18931zn3.
(PDF)

**S9 Fig. Estimates of mean (± 95% CI) excess ancestry heterozygosity for individuals and loci across "replicates".** We consider a replicate to be a unique biparental $F_0$ cross for aquarium studies and a unique pond for pond studies. Mean excess ancestry heterozygosity is shown for each such replicate for both individuals (upper) and loci (lower). In each panel, the horizontal line indicates no excess ancestry heterozygosity. Red points are "lab" replicates, and blue points are "pond" replicates. The data and code required to recreate this figure may be found at https://doi.org/10.5061/dryad.h18931zn3.
(PDF)

**S10 Fig. de Finetti ternary diagrams for genotyped loci.** Each point represents genotyped locus within a given study (i.e., line in Table 1 in the main text) and shows the frequency of either benthic or marine alleles on the x-axis and its heterozygosity on the y-axis. These graphs are not used for analysis, but rather are shown to allow readers to visualize the structure of the raw data that underlies our analysis. Specifically, the shapes of the distributions of heterozygosity and hybrid index values are similar between environments and crosses—the means are just subtly different. The data and code required to recreate this figure may be found at https://doi.org/10.5061/dryad.h18931zn3.
(PDF)

**S11 Fig. Main analysis with observed ancestry heterozygosity rather than excess ancestry heterozygosity as the response.** For full details, see caption of Fig 2 in the main text. Qualitative conclusions of statistical models are identical to those of the main analysis (see archived R script). The data and code required to recreate this figure may be found at https://doi.org/10.5061/dryad.h18931zn3.
(PDF)

**S12 Fig. Test of main hypothesis with loci instead of individuals.** Figure is as in Fig 2 in the main text (and S11 Fig) and qualitative conclusions of statistical models are identical to those of the main analysis (see archived R script). The data and code required to recreate this figure may be found at https://doi.org/10.5061/dryad.h18931zn3.
(PDF)

## Acknowledgments

Feedback from D. Irwin, S. Otto, L. Rieseberg, and R. Stelkens improved the manuscript. Discussions with the Schluter Lab at the University of British Columbia and the Schumer Lab at Stanford University improved the analysis. We are grateful to the authors of the primary studies from which we gathered data for their data stewardship.

## Author Contributions

**Conceptualization:** Ken A. Thompson, Molly Schumer, Dolph Schluter.

**Data curation:** Ken A. Thompson, Catherine L. Peichel, Diana J. Rennison, Arianne Y. K. Albert, Timothy H. Vines, Anna K. Greenwood, Abigail R. Wark, Dolph Schluter.

**Formal analysis:** Ken A. Thompson, Yaniv Brandvain, Molly Schumer, Dolph Schluter.

**Funding acquisition:** Catherine L. Peichel, Dolph Schluter.

**Investigation:** Ken A. Thompson, Catherine L. Peichel, Diana J. Rennison, Matthew D. McGee, Arianne Y. K. Albert, Timothy H. Vines, Anna K. Greenwood, Abigail R. Wark, Yaniv Brandvain, Molly Schumer, Dolph Schluter.

**Methodology:** Ken A. Thompson, Catherine L. Peichel, Diana J. Rennison, Matthew D. McGee, Arianne Y. K. Albert, Timothy H. Vines, Anna K. Greenwood, Abigail R. Wark, Molly Schumer, Dolph Schluter.

**Project administration:** Ken A. Thompson, Catherine L. Peichel, Dolph Schluter.

**Resources:** Catherine L. Peichel, Dolph Schluter.

**Supervision:** Catherine L. Peichel, Molly Schumer, Dolph Schluter.

**Visualization:** Ken A. Thompson, Matthew D. McGee, Dolph Schluter.

**Writing – original draft:** Ken A. Thompson, Dolph Schluter.

**Writing – review & editing:** Ken A. Thompson, Catherine L. Peichel, Diana J. Rennison, Matthew D. McGee, Arianne Y. K. Albert, Timothy H. Vines, Anna K. Greenwood, Abigail R. Wark, Yaniv Brandvain, Molly Schumer, Dolph Schluter.

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
