## [Editor Report · Decision Letter 0]

15 Jul 2021

Dear Dr Thompson, 

Thank you for submitting your manuscript entitled "Genetic evidence for environment-dependent hybrid incompatibilities in threespine stickleback" for consideration as a Short Report by PLOS Biology.

Your manuscript has now been evaluated by the PLOS Biology editorial staff and I am writing to let you know that we would like to send your submission out for external peer review.

Please accept my apologies for the delay incurred while we sought external advice; because of Academic Editor absences, we were unable to secure advice on your paper, and so we will be looking for some enthusiasm from the reviewers to judge whether you paper is ultimately better suited to PLOS Biology or PLOS Genetics.

Kind regards,

Roli Roberts

Roland Roberts

Senior Editor

PLOS Biology

rroberts@plos.org

---

## [Decision Letter · Decision Letter 1]

24 Aug 2021

Dear Dr Thompson,

Thank you very much for submitting your manuscript "Genetic evidence for environment-dependent hybrid incompatibilities in threespine stickleback" for consideration as a Short Reports at PLOS Biology. Your manuscript has been evaluated by the PLOS Biology editors, an Academic Editor with relevant expertise, and by three independent reviewers.

In light of the reviews (below), we are pleased to offer you the opportunity to address the comments from the reviewers in a revised version that we anticipate should not take you very long. We will then assess your revised manuscript and your response to the reviewers' comments and we may consult the reviewers again.

IMPORTANT: You'll see that the reviewers are all very positive about your study, but each has a number of requests for improvement. The Academic Editor supplied the following additional advice that you might find useful when prioritising your revisions: "I think reviewer #2 raises some interesting points about interpretation that need to be addressed. In particular his/her suggestion that inbreeding might lead to similar patterns seems valid to me - I think the authors need to address this in the discussion. I would be less concerned about some of the points raised by reviewer #3, such as using a different set of genetic markers, which would represent a significant additional burden of work. I think this goes beyond the scope of this paper, but the implications of possible biases in RAD-seq should be discussed."

We expect to receive your revised manuscript within 1 month.

**IMPORTANT - SUBMITTING YOUR REVISION**

*Resubmission Checklist*

*Published Peer Review*

*PLOS Data Policy*

*Blot and Gel Data Policy*

Sincerely,

Roli Roberts

Roland Roberts

Senior Editor

PLOS Biology

rroberts@plos.org

REVIEWERS' COMMENTS:

Reviewer #1:

I really appreciated this paper addressing the interesting question of coarse effects of incompatibilities across the genome. 

I found it concise and well written.

I like the approach and the use of the existing data on sticklebacks to explore the interplay between incompatibilities and the environment.

I only have a few minor questions/comments.

L73: "relatively homozygous" -> relatively more homozygous?

L98 "interactions between individual hybrids": do you mean competition and/or sexual selection? It might be worth adding some detail. 

Additionally, biotic interactions can also happen with parents don't you think?

I do not grasp here what you want to convey by separating individual hybrids from other abiotic and biotic factors.

L339 about yeast: I am not a yeast expert at all, but I remember there can be frequent aneuploidies in some crosses. Is it possible it could explain this pattern of low heterozygosity if parts of the genome end up in one copy?

L348: How does ponds and lake environments differ in all those stickleback experiments? 

While the difference compared to marine environments for example might not be too high, 

I expect the environment of a small pond might still be quite different from the one of a big lake. 

In such cases there might be some room for positive selection of some transgressive phenotypes.

Reviewer #2:

In their manuscript titled "Genetic evidence for environment-dependent hybrid incompatibilities in threespine stickleback" Thompson et al. explore the genomic evidence for sets of trait mismatches in hybrids that reduce fitness (what the authors refer to as 'ecological incompatibilities'). This paper presents a creative follow-up to a previously published meta-analysis of trait distributions (and their potential fitness implications) in hybrids (Thompson et al. 2021), and integrates (mostly) previously published genetic/genomic datasets to assess if hybrids reared in semi-wild conditions experience selection against mixed ancestry and subsequent increased heterozygosity more so than hybrids reared in the lab.

Overall I thoroughly enjoyed the manuscript and feel this work provides much needed perspective. I also applaud the authors for a creative use of previously published data. The work makes a novel and, in my opinion, very important contribution to the speciation literature. It is also well written and extremely thorough in analysis. I applaud the authors (again!) for the transparency of the data and methodology. Although my review is overall very positive, I do have some concerns that I think could be addressed with written caveats or minor additional analyses.

1. The authors make the argument that environment-dependent heterosis is unlikely to explain the small increase in heterozygosity in pond populations versus lab reared populations of stickleback hybrids (lines 278-327), although they do acknowledge that "Ultimately, the data presented here have limited ability to conclusively distinguish between single-locus processes like heterosis and multi-locus processes like incompatibilities" (lines 281-284). While I mainly find their arguments against environment-dependent heterosis compelling, there is one scenario that I think is unaccounted for in the current manuscript, which is environment-specific inbreeding depression (given full sibling mating from wild collected grandparents). For example, imagine a scenario in which one or both wild-caught parents harbors recessive deleterious alleles in heterozygous form (perhaps at many loci). Given that these alleles are predicted to be relatively common overall, but rare for a given locus, F1s are also likely to be heterozygous for any given recessive deleterious allele. However, if full sibling F1s are mated, then F2s have a high probability of inheriting both recessive deleterious alleles, and expressing some manifestation of inbreeding depression. In this case, the lines of evidence against environment-specific heterosis are not as compelling: (1) we do not expect that F1s should show heterosis/ increased fitness (recessive deleterious alleles are equally masked in both F1s and in wild parents). (2) Condition might not correlate with overall heterozygosity (just heterozygosity at specific regions of the genome) and/or the inbreeding depression in question might not affect overall condition. (3) I find the argument that excess heterozygosity should decline with further deviations from a hybrid index of 0.5 (and the data presented) fairly weak and unconvincing overall, particularly as these models are indistinguishable when accounting for the vastly different sample size via bootstrapping. Furthermore, if there is any epistatic variation for environment-specific inbreeding depression, one might expect the same results as under a model of 'ecological incompatibilities'. (4) The data from re-analyzed from Arnegard et al. is quite compelling, although I think this particular argument might benefit from subsampling the 'not A' ground to be the same size as the 'A' group, and asking how often these two classes differ from each other (and from 0). 

2. This is not an issue per se, but it seems the authors are really wanting to detect whether hybrids have a dearth of pairs of alleles with alternate homozygous ancestry, with an increase in heterozygosity being a natural side effect of this. I wonder if a complimentary analysis would be to calculate some modified hybrid index using only loci that are homozygous within an individual and ask if surviving individuals are skewed to one parent or the other at homozygous regions in the pond versus lab. To avoid any issues caused by directional selection, the authors would need to find skew in both directions (i.e. for both parents) within their dataset(s). I think this type of analysis might also quell some worries about the role of inbreeding depression (and other explanations for environment-specific heterosis) in their dataset, if it's possible to do.

3. I think the method presented by the authors is, overall, of great value to the field, and as previously stated, this work has creatively re-used previously collected data to say something very new, and in my mind, of utmost importance. However, while the authors elucidate throughout the manuscript that ability to detect differences in heterozygosity between field vs lab reared hybrids at these genome-wide coarse scales will depend largely on the polygenic nature of the traits of interest and the scale of additivity/dominance for the alleles controlling those traits. I think the authors should perhaps highlight this caveat under the 'Outlook, caveats, and conclusions' section.

I have many, relatively minor and surely picky points that I think would improve the clarity of the manuscript:

- Lines 43-45: For the severity of 'intrinsic' incompatibilities varying across environments, I also suggest referencing Bundus et al. 2015. Gametic selection, developmental trajectories, and extrinsic heterogeneity in Haldane's Rule. Evolution. 

- Line 46: "the number of intrinsic incompatibilities" feels ambiguous to me, because it is unclear whether the authors are referring to the number of interacting loci (as the references would suggest) or the number of unique incompatibilities (which may or may not influence the same hybrid phenotype). 

- Line 48: I would also reference Chae et al. 2014. Species-wide genetic incompatibility analysis identifies immune genes as hot spots of deleterious epistasis. Cell. 

- Lines 51-55: I might include the example of extreme transgressive trait combinations and developmental mismatches outlined in Coughlan et al. 2021. The genetic architecture and evolution of life history divergence among perennials in the Mimulus guttatus species complex. Proc B.

- Lines 59-60: I find this sentence a bit odd, as the discussion of how polygenic traits are depends, in part, on what type of selection is acting on them. While there is good reason (and evidence) to suggest that Rockman's 2012 piece describes quantitative traits that under stabilizing selection within populations (or even within species), there are also good theoretical reasons to think that the genetic architecture of divergence should be much less complex (see Remington 2015. Alleles versus mutations: Understanding the evolution of genetic architecture requires a molecular perspective on allelic origins. Evolution). Of course, the dearth of genetic studies quantifying the genetic architecture of species divergence and its relation to fitness make these two world views somewhat difficult to assess, but to date there are many studies which have shown that species divergence can be explained by a fairly small handful of loci. This has implications for how effective this differential heterozygosity test will be (see above).

- Lines 210: is this really variation in recombination, or is it variation in independent assortment of chromosomes and/or random chance in which product of meiosis is destined to be the egg (in the case of female meiosis). 

- Lines 214-216: Could the authors provide a supplemental figure with the raw, observed heterozygosity data plotted? Similarly, could the authors add the data from the three lab studies and Schluter et al. 2021 to Figure S3? I think this would better allow readers to look at the variance among lab studies and marine studies as well. (The authors could even add each as a separate facet/panel so as to avoid confusion by comparing between different crosses and/or environments). 

- Lines 218-230 (Data Analysis section)- do the authors think it would be beneficial to do some sort of bootstrapping of the pond data so that sample sizes are matched between pond and aquaria (which overall have much lower sample sizes) to see how often pond heterozygosity is statistically elevated from 0?

- Throughout the manuscript the authors suggest that mismatched homozygous ancestry is resulting in trait mismatches (which in turn have fitness consequences). Although the some of the authors have previously shown the importance of trait mismatches in hybrid fitness (for example, Thompson et al. 2021), I would be cautious to so definitely say that the reduction in fitness is due explicitly to trait mismatches vs transgressive values of single traits (due to epistatic interactions, for example). 

- I just want to reiterate that the link to phenotype, as described in lines 317-327 is absolutely lovely- huge congrats to the authors for thinking of this.

Lines 339-342: the authors might also find the following reference useful: Smukowski Heil et al. 2019. Temperature preference can bias parental genome retention during hybrid evoluition. PLoS genetics.

Reviewer #3:

The authors present a new analysis of numerous previous pond and lab experiments measuring the survival of stickleback hybrid crosses. They make the fascinating observation that heterozygosity is about 3% higher in pond-raised stickleback than lab-raised stickleback in both benthic x limnetic crosses and marine x freshwater crosses. They conclude that this pattern is consistent with extrinsic genetic incompatibilities, i.e. field selection against incompatible interacting homozygous loci from different parental lines which does not occur in the lab environment. They reject the hypothesis that heterosis in field environments better explains the observed pattern, i.e. hybrids with more heterozygous loci are more fit in ponds. 

I think this is a fascinating observation that will inspire numerous field biologists to conduct similar studies or re-examine their old data to look for this pattern. It is also open to numerous interpretations - the strongest alternative being heterosis/balancing selection. The authors are careful to discuss this alternative explanation that heterozygous loci are directly selected for in field environments and offer several arguments in favor of genetic incompatibilities. Indeed, their conclusion is careful in stating that the observed heterozygosity "is consistent with" extrinsic genetic incompatibilities. I agree that it is consistent, but I think the authors could be more balanced throughout their presentation - particularly in the title, abstract, and introduction. In my view, they do not present explicit evidence for extrinsic genetic incompatibilities - the pattern could also be explained by simple heterosis in field environments. 

The analyses presented are relatively short, although they do explore some simulations. I was also left wondering - how many genetic incompatibilities should we expect in these crosses and are they enough to be predominant in the genome-wide signal of 3% increased heterozygosity? The incredible replication across different crosses, genotyping designs, and studies is impressive and some of these supplemental results should be included in the main manuscript if possible. 

I would expect many more incompatibilities in the more divergent marine x freshwater crosses than within-lake benthic x limnetic crosses, particularly given the snowball effect, however excess heterozygosity doesn't appear to increase in more divergent crosses? Why not? 

I also think the authors could compare different types of heterozygous sites to better test their hypotheses. Is excess heterozygosity still observed in gene desert regions or only genic/near-gene regions? Do gene regulatory regions and/or coding regions exhibit more excess heterozygosity? The authors would have to postulate additional reasons for different site patterns between heterosis and genetic incompatibilities, but this could still point in useful directions for sorting alternative explanations. 

Finally, all the pond hybrids were the result of natural assortative mating events in the field whereas all the lab hybrids were artificial crosses? As I note below, are the authors worried about the effects of increased inter-F1 family assortative mating resulting in increased heterozygosity in ponds? More detail and controls for lab artificial crossing schemes would be useful. 

Similarly, models exploring the effect of different genetic markers on heterozygosity would also be useful. RADseq has a well known bias against detection of heterozygous sites due to restriction-site dropout (Arnold et al. 2013). 

Line 26: Could the authors qualify this a bit more? I suppose they don't consider tests of postzygotic extrinsic isolation to be tests of hybrid incompatibilities?

Line 32: Space is limited, but it would be good to know why heterosis is an unlikely explanation? The authors could be much more balanced throughout in presenting heterosis as a plausible additional explanation.

Also, I think the authors should clarify in line 30 that this excess heterozygosity could also be a signature of heterosis, not just selection against incompatibilities as they claim, correct?

Line 35: incompatibilities or environment-dependent heterosis?

Line 76: How does this scale with the genetic architecture of the phenotype? What happens for more polygenic traits if opposite homozygous ancestry is not the predominant pattern?

Line 88: The authors are also assuming some disruptive selection on trait-matched hybrids to get the saddle shape? As stated, this suggests a ridge, not a saddle.

Line 97: I'm not convinced this is 'compelling' support for extrinsic incompatibilities because there are alternative explanations for the observed pattern of increased heterozygosity in field samples. 

Line 126: This looks thorough, but can the authors also run a glm with the effects of experiment type, genotyping method, and their interaction on heterozygosity levels? It would be good to know if the magnitude of increased heterozygosity observed in the field environments differs across different studies?

Line 136: So, all field experiment hybrids were the result of natural assortative mating whereas all lab 'control' hybrids were random crosses with no assortative mating? Since there are multiple F1 families in each experiment, was there increased inter-family assortative mating?

I agree that only considering fixed differences between parents in the F2 generation probably makes this a moot point, but could assortative mating also potentially explain an increase in the survival of more heterozygous fish in fish enclosures? E.g. increased assortative pairings between F1 families in pond experiments could result in more outbred F2 offspring compared to random lab pairings?

Line 222: An additional model should include the effect of genotyping method (GBS, etc.). Could also include models with effects of sex, presence/absence cutthroat trout, assortative mating/or artificial crosses in the lab to see which model best explains heterozygosity patterns?

Line 295: Is this true - there was zero mortality in the lab before juveniles were released into field enclosures? Heterosis could also act on increased viability of eggs - were there zero egg deaths in lab reared clutches?

Line 305: remove extra 'which'

Line 312: This is a very interesting argument in support of the author's extrinsic genetic incompatibilities argument. Should we really expect the fitness effects of genetic incompatibilities to decline linearly from 0.5? Also, why isn't a quadratic fit used as fitness should decline in either direction away from 0.5 ancestry? 

Line 367: Citations would be useful here, too?

Fig. 1 could probably be expanded with at least on additional panel. E.g. what are the effects of selection on trait mismatch and heterozygosity? How does heterozygosity relate to fitness effects of genetic incompatibilities vs. heterosis (see my previous comment)?

Looking at Fig. 2, I wonder if this pattern is mostly/entirely driven by the tails of the distribution in pond experiments? Would ranked analyses produce similar results to parametric results? 

Some fish are approaching 40% heterozygous ancestry? What is the predicted vs. observed frequency of these individuals in an F1 intercross (the design of these studies)?

---

## [Editor Report · Decision Letter 2]

1 Nov 2021

Dear Dr Thompson,

Thank you for submitting your revised Short Reports entitled "Genetic data suggest that hybrid incompatibilities in threespine stickleback are environment-dependent" for publication in PLOS Biology. The Academic Editor and I have assessed your revisions and responses to the reviewers. 

Based on this assessment, we will probably accept this manuscript for publication, provided you satisfactorily address the following editorial and policy-related requests.

IMPORTANT:

a) We suggest something more specific for the title: perhaps "Genetic signatures suggest that hybrid incompatibilities in threespine stickleback are environment-dependent" or "Ancestry heterozygosity suggests that hybrid incompatibilities in threespine stickleback are environment-dependent" or "Analysis of ancestry heterozygosity suggests that hybrid incompatibilities in threespine stickleback are environment-dependent" 

b) Your financial statement currently says “The authors received no specific funding for this work.” Can you confirm that this is indeed correct?

c) Please address my Data Policy requests below; specifically, please supply numerical values and/or the code required to recreate Figs 1AB, 2AB, S1, S2, S3, S4, S5, S6, S7, S8 S9, S10, S11, S12, and cite the location of the data clearly in each relevant Fig legend.

We expect to receive your revised manuscript within two weeks. 

*Published Peer Review History*

*Early Version*

Sincerely,

Roli Roberts

Senior Editor,

rroberts@plos.org,

PLOS Biology

DATA POLICY:

We note that the raw data are provided in your supplementary data files. However, we also need the individual numerical values that underlie the figures to be made available in one of the following forms:

Regardless of the method selected, please ensure that you provide the individual numerical values that underlie the summary data displayed in the following figure panels as they are essential for readers to assess your analysis and to reproduce it: Figs 1AB, 2AB, S1, S2, S3, S4, S5, S6, S7, S8 S9, S10, S11, S12. NOTE: the numerical data provided should include all replicates AND the way in which the plotted mean and errors were derived (it should not present only the mean/average values).

DATA NOT SHOWN?

---

## [Editor Report · Decision Letter 3]

4 Nov 2021

Dear Ken,

On behalf of my colleagues and the Academic Editor, Chris Jiggins, I'm pleased to say that we can in principle accept your Short Reports "Analysis of ancestry heterozygosity suggests that hybrid incompatibilities in threespine stickleback are environment-dependent" for publication in PLOS Biology, provided you address any remaining formatting and reporting issues. These will be detailed in an email that will follow this letter and that you will usually receive within 2-3 business days, during which time no action is required from you. Please note that we will not be able to formally accept your manuscript and schedule it for publication until you have any requested changes.

IMPORTANT:

a) Many thanks for supplying the Dryad link. This enabled me to check it and sign off on the MS editorially. However, you will need to include the Dryad URL/DOI in each relevant main and supplementary figure legend (e.g. "The data and code required to recreate this Figure may be found at https://datadryad.org...."). I've left a note with my Production colleagues to contact you about this.

b) Those fish pics in Fig 2 are fantastic!

PRESS: We frequently collaborate with press offices. If your institution or institutions have a press office, please notify them about your upcoming paper at this point, to enable them to help maximise its impact. If the press office is planning to promote your findings, we would be grateful if they could coordinate with biologypress@plos.org. If you have not yet opted out of the early version process, we ask that you notify us immediately of any press plans so that we may do so on your behalf.

Best wishes,

Roli 

Roland G Roberts, PhD 

Senior Editor 

PLOS Biology

rroberts@plos.org